

# Landfalls of Tropical Cyclones with Rapid Intensification in the Western North Pacific

Jie Yang[1,2] and Meixiang Chen[3]

[1]Key Laboratory of Coastal Disaster and Defence of Ministry of Education, Hohai University, Nanjing, 210024, China
[2]Department of Civil and Environmental Engineering, National University of Singapore, 117576, Singapore
[3]College of Oceanography, Hohai University, Nanjing, 210024, China

*Correspondence to*: Jie Yang (jie_yang@hhu.edu.cn) and Meixiang Chen (chenmeixiang@hhu.edu.cn)

**Abstract.** This study statistically investigates the seasonal and decadal variation of tropical cyclones (TCs) underwent rapid
intensification (RI) and their landfalling cyclone energy in the western North Pacific using the satellite-era best track data
from 1986 to 2017. Totally 31.2 % TCs have underwent at least one RI processes (RI TCs) and 341 made landfalls after RI
from 946 historical TCs, using the definition of 95th percentile from the accumulative probability distribution of over-water
24-h TC intensity change. The frequent-occurrence region of RI is found in sea areas to the east of Philippines, and the mean
genesis and on-set locations of landfalling TCs underwent RI had westward components compared with the ones did not
made landfalls. The Philippine coast, the southern Chinese coast and the coast along the southern Japan are the three main
regions affected by the landfalling RI TCs. The coasts in the latter two regions have increased trend of cyclone energy since
1986, which possibly correlates with the poleward migration of the mean latitude where TCs reach their lifetime maximum
intensities (LMI). The frequency of the landfalling RI TCs have a significant upward trend with insignificant increase in
their LMI, while both the LMI and landfalling cyclone energy by TCs that didn't undergo RI in the western North Pacific
show downward trends in the period during 1986–2017. The changes of the LMI distribution in the western North Pacific are
related tightly with these two types of TCs with different intensification rates: strong TCs are found become stronger mainly
due to more active RI processes, while weak TCs have weakened in majority of moderate intensity TCs, which didn't
experience RI processes.

## 1 Introduction

Tropical cyclones are one of the most devastating natural hazards, their activity of landfalls and associated landfalling
intensities (or energy) are highly related to coastal damages and have profound social and economic impacts around the
world. Among the TCs, the ones intensify rapidly challenge the scientists with the timely accurate intensity forecast, and
historically most of them developed into high intensity storms (e.g., major hurricanes on the Saffir–Simpson Scale), as
observed in all basins (Lee et al., 2016). Moreover, the TCs that undergo rapid intensification (RI) just prior to their landfalls





are classified into one of the most dangerous types and can lead to disastrous scenarios in coastal communities with inadequate preparation and evacuation efforts (e.g., Typhoon Hato (2017) and Hurricane Harvey (2017)). Due to the fact of the observed sea level rise and land subsidence, as well as the booming population and economy in coastal regions, the TCs approaching shore and making landfalls tend to pose more threat to coasts with increasing exposure. Although several studies proved that social factors (e.g., increase in population and wealth) has played a major role in coastal damages prone

to landfalling TCs (Klotzbach et al., 2018; Weinkle et al., 2012; Zhang et al., 2009), the changes in storminess (i.e., the frequency and intensity) from historical records and the trend detection and attribution of TC activities under future climate change, are of great concern by previous research (Elsner et al., 2008; Emanuel, 2005; Knutson et al., 2010; Mei and Xie, 2016; Webster et al., 2005) and are still crucial for further vulnerability assessment, preparation and mitigation of coastal regions. However, for the western North Pacific basin, where on average more than one-third of global TCs were generated

and has extensive TC activities (Mei et al., 2015), the trend of TC landfalls and the cyclone energy accumulated by TCs with different intensification rates are not fully examined.

The locations where TCs make landfalls are largely dependent on their tracks (Zhou et al., 2018), which can be classified into three prevailing directions over the western North Pacific: westward-moving, recurving-land, and recurving-ocean (Colbert et al., 2015; Elsner and Liu, 2003; Wu and Wang, 2004). The meteorological factors controlling or contributing to

TC tracks, e.g., the large scale steering flow (Zhou et al., 2018), affect the resultant TC landfall locations. Decay in wind speed due to land-sea interaction and asymmetric structure are ones of mostly focused topics associated with TC landfalls (Chan et al., 2004; Chan and Liang, 2003; Chen and Yau, 2003). Meanwhile, the studies with emphasis on track density variability (Mei et al., 2015), TC recurvature (Zhang et al., 2013b), TC-land interaction during landfalling processes (Zhang et al., 2013a) have gained knowledge about TC tracks and landfall related mechanisms. Mei and Xie (2016) suggested the

landfalling typhoons in the northwest Pacific have significantly intensified since the late 1970s owing to strengthened intensification rates, by investigating four clusters of typhoons with different track characteristics. Note that the TC intensity prior to landfall is not necessarily correlated to the TC peak intensity, extra metric is necessary to better describe the landfalling cyclone energy. Although debates still exists that if TC frequency has an upward trend or not, researchers have gained agreement to some extent that the numbers of high intensity TCs indicate increased trend in the past decades (Song et

al., 2018; Webster et al., 2005). Lee et al. (2016) has demonstrated that the TCs underwent RI and the TCs didn't undergo RI (non-RI TCs) are main storm types contributing to major storms and minor storms, respectively. Thus, their separate variation of frequency and intensity, the associated influence on the TC activities, as well as the spatial distribution of the landfalling TCs along regional coasts are the main concern in the present study.

In this paper, we examine landfalls in the western North Pacific by TCs with different intensification rates, namely RI TCs

and non-RI TCs, using the IBTrACS data set (Knapp et al., 2010); one straightforward metric was proposed to assess the storm power of the landfalling TCs. The rest of the manuscript is arranged in the following sequence: Section 2 explains the utilized data sets, definition and methodology applied in the subsequent sections. In section 3, the LMI distribution of the TCs is explored and we detect all the landfalls during the period 1986–2017 and examine their destructive power with a





metric describing their landfalling cyclone energy, for RI TCs and non-RI TCs, respectively. Focusing on the decadal and
seasonal variability of TCs, several characteristics, e.g., LMI, accumulated landfalling cyclone energy, TC genesis and RI
on-set locations, etc., were investigated in section 4 and section 5. Finally, a concluding summary is provided in section 6.

## 2 Data and Methodology

The latest released version v04r00 of IBTrACS best track data (Knapp et al., 2010; https://www.ncdc.noaa.gov/ibtracs/)
developed by NOAA's National Climatic Data Center was used in the present study. The best track data includes geography

location, maximum sustained winds (MSW) and minimum atmospheric pressure, as well as extra information, e.g., the
radius of maximum wind (RMW), while available from the agencies as provided. The locations are provided with the spatial
resolution of 0.1°×0.1°, the magnitudes of 1-min MSW are rounded to nearest 5 kt. Compared with previous versions, the
track data of version v04r00 is available in every 3 h, when the locations and the surface wind speed are interpolated from
the data provided by agencies using the B-spline and the linear interpolation, respectively. For simplicity and to avoid the

possible uncertainty introduced by interpolation, here we use the 6-h track data by excluding the interpolated track locations.
The data provided by the Joint Typhoon Warning Center (JTWC) was utilized from the data set (IBTrACS-JTWC herein for
short) for analysis in the present work. Dvorak's satellite-image-based technique (Dvorak, 1984) was used after 1985 by
almost all the agencies around the world to estimate cyclone intensity and has enhanced the quality of observations among
basins [Knaff et al., 2010]. We utilized the data during the period from 1986 to 2017 in the present study for a better

consistency of data quality. The TCs with keyword 'TRACK_TYPE' as 'main' (of which the data has been re-analyzed and
has higher quality) and with track data available were counted into track samples for analysis. By further excluding the ones
with lifetime less than 24 hours, the total number of TC track samples, being involved in the present study, counts 946 for
the western North Pacific during 1986–2017.

The rate of intensity change in 24 h, notated as $\Delta V_{24}$, was calculated using the wind speed change for each of the track

segments within temporal span of 24 h. The track segments of TCs over land and extratropical were excluded for $\Delta V_{24}$
calculation. The rapid intensification of TCs adopts a certain criterion of $\Delta V_{24}$, a fixed magnitude of $\Delta V_{24}$ and a fixed level in
the percentile of $\Delta V_{24}$ cumulative distribution are the most accepted ones (Kaplan and DeMaria, 2003; Wang and Zhou,
2008). Nevertheless, among different basins, the criterions of RI magnitude associated with the same percentile have
discrepancy (Sampson et al., 2011; Wang et al., 2017), resulting in inconsistency of statistic characteristics from the RI

probabilities in different basins. Kaplan and DeMaria (2003) found that TCs with RI larger than 30 kt in the Atlantic basin is
approximately equivalent to the 95[th] percentile of $\Delta V_{24}$ cumulative distribution from 1989 to 2000. Using the IBTrACS-
JTWC track data during period 1986–2017, the accumulative distribution of over-water $\Delta V_{24}$ in the western North Pacific is
shown in Fig. 1. The result indicates that $\Delta V_{24}$ larger than 30 kt is only equivalent to 92[th] percentile in the western North
Pacific. Lee et al. (2016) also reported that 35 kt is better to distinguish the TCs with RI and TCs without RI in most of the

basins. To that end, the RI of TCs in our study is defined with over-water $\Delta V_{24}$ larger than 35 kt of 1-min MSW, which is



equivalent to 95$^{th}$ percentile of cumulative distribution (the corresponding RI definitions for CMA/JMA best track data could also be found in Appendix A). Similar as the classification of intensity change groups by Fischer et al. (2018), we develop three groups to further distinguish TCs with different maximum intensification rates during their strengthening processes in lifetime (Table 1). The additive of slow-intensified (SI) TCs and neutral (N) TCs constitutes the non-RI TCs.

To quantify the power of the landfalling TCs and to obtain associated spatial distribution along the coast, control lines (CTRLINE hereafter; see Fig. 2a) in this region were created and split into totally 106 segments with separate length coverage of ~200 km. The landfalls were detected firstly by automatically judging from the keyword 'DIST2LAND' in the IBTrACS-JTWC data, then were final-confirmed via separately visual check. According to the detected landfalls during 1986–2017 in the western North Pacific shown in Fig. 2b, the most affected regions are the Philippine coast and the Vietnam

& the southern Chinese coast by mainly the westward TCs, as well as the region in relative higher latitude (> 30˚ N; including the coast along the northern China, Korean Peninsula and Japan) by mainly northeastward recurving TCs. Inspired by the accumulated cyclone energy proposed by Knapp et al. (2013), which described the total energy carried during a TC's life span, we develop the metric $MACE$, multidecadal accumulated cyclone energy, to investigate the intensity changes in landfalling activities. For a specific period, the value of $MACE$ was calculated for each segment on CTRLINE as the sum of

the square of the maximum sustained wind prior to landfall, and was normalized in the form:

$$MACE = \frac{1}{T}\sum_{i=1}^{N_{TC}} MSW_i^2 \qquad\qquad (1)$$

where $N_{TC}$ is the number of TCs that made landfalls at one CTRLINE segment for the specific period, $MSW_i$ is the maximum sustained wind just before TC's landfall (at 6-h resolution), $T$ is the number of years for $MACE$ calculation. Therefore, $MACE$ is an annual averaged metric.

## 115   3 LMI Distributions of RI and Non-RI TCs and Their Landfalls

### 3.1 RI in the western North Pacific

Figure 3a represents the locations of totally 953 RI processes in the western North Pacific, 50 kt and 70 kt of 1-min MSW were used to further distinguish grades of RI. The majority of RI ranges 35–50 kt, only 23.2% and 5.9% of them could reach up to 50 kt and 70 kt, respectively. At the same time, the proportion of RI duration within 24–48 h was up to 65.9%, while a

percentage of 27.1% had longer RI processes for 42–72 h, and only 7.0% lasted more than 72 h without exceeding 102 h. The results indicate 295 TCs (31.2 %) had underwent at least one RI during their lifetime, namely 9 TCs per year on average. This ratio is equivalent to that of the Atlantic basin (31%; Kaplan and DeMaria, 2003). The distribution of RI occurrence density by 1˚×1˚ grid (Fig. 3b) illustrates that the main region undergoing the most frequent RI was the sea areas east of the Philippine Islands, namely the region 10˚ N–20˚ N, 124˚ E–137˚ E.



## 3.2 LMI Distributions of RI and non-RI TCs

Among all the RI TCs, 192 of them made at least one landfall finally, which is up to 65.1%. As TC landfall intensity is largely affected by its life cycle and LMI, the LMI distribution of TCs is investigated before looking into the intensity prior to their landfalls. The LMI distribution for all TCs in the western North Pacific in Fig. 4a shows bimodal pattern with $1^{st}$ and $2^{nd}$ peaks at about 30 kt and 140 kt, which are highly contributed by non-RI TCs (Fig. 4c–d) and RI TCs (Fig. 4b), respectively. The LMI distribution of non-RI TCs is mainly depend on 'SI' group of TCs, only 71 TCs are observed in the 'N' group and are mainly comprised with tropical depressions. These characteristics are consistent with Lee et al. (2016)'s conclusion that the bimodal pattern of LIM distribution are satisfied in all basins, and the non-RI TCs and RI TCs contribute mainly to the low and the high intensity TCs, respectively. Note that of all the major storms (Category 3–Category 5), the TCs underwent RI take the percentage of 87.4%, and reach up to 91.9% of high intensity storms (Category 4–Category 5). The ratio is even hundred percent for the Category 4 and Category 5 storms in the Atlantic basin that experienced RI (Wang and Zhou, 2008). Above statistics indicate that the majority of major storms underwent at least one RI process, and remind us the significance of RI, being one of the essential characteristics for the high intensity TCs. Among the major storms, the ones approached the coast undergoing RI processes and finally made landfalls pose a great threat to coastal communities, especially which lack evacuation preparation.

## 3.3 Annual Averaged Landfalling Intensity of TCs

From all detected landfalling TCs, 341 landfalls from 192 RI TCs and 587 landfalls from 357 non-RI TCs were counted during the period 1986–2017 (Fig. 5a–b). Even that the number of non-RI TC landfall occurrence far overrode that of RI TCs, namely nearly 2 times, we observe their landfalling intensities being mostly weaker than Category 1, except a few TCs that made landfalls along the coast of Zhejiang Province and Taiwan, and of southern Japan, with intensity above Category 3. On the opposite side, RI TCs had affected the coast along both Philippines and southern China with landfalling TCs reached Category 4, or even Category 5. Utilizing the concept $MACE$, we estimate annual average values of landfalling energy along the coast in the western North Pacific from both types of TCs. This energy was further transformed to counts in four intensity grades as in Table 3. The criteria of grade II–IV are decided using the lower criteria of tropical storms, Category 1 storms and Category 4 storms in sequence on the Saffir–Simpson hurricane wind scale, representing energy level associated with tropical storms, moderate and high intensity storms, respectively. The accumulated energy is classified into 1–n counts of criteria $MACE$ units, before it could be categorized into next higher level of grade. To have an example for illustration, $2\ CE_2$ at a specific segment of coastline demonstrates that the energy reaches at least double of $CE_2$ and is less than trible of $CE_2$, and so on for counts in other grades. Using this quantitative method, the annual averaged statistics of cyclone energy in Fig. 5c shows: the Philippine islands and Taiwan island were the most affected regions by landfalling TCs, at least one storm equivalent to Category 1 and above hit these regions each year; the southern Japanese coast, the southern Chinese coast and the northern Vietnam coast were influenced by lower grade of cyclone power. Comparatively, the power





brought by RI TCs was more compact and denser on the Philippine coast (Fig. 5d), while the non-RI TCs tend to have more evenly-distributed energy, affecting both the Philippine coast and the southern Chinese coast (Fig. 5e). In view of greater potential impacts of RI TCs on coastal region, their seasonal and decadal characteristics are necessarily investigated and to

be illustrated in next sections.

## 4 Seasonal Variation of RI

### 4.1 Origin and On-set Locations of RI TCs

During the cyclone active season, the majority of the landfalling RI TCs occurred until the late autumn, their accumulated number maintains at a ratio above 20%, and lasted from July to November. September is the most active month throughout

the year, with maximum numbers of both total TCs and the landfalling RI TCs. Taking May, June and December into consideration as well, the monthly averaged geography locations of RI TC genesis and on-set were calculated using TC samples during 1986–2017 (Fig. 7); among them, the corresponding representative locations associated with the landfalling RI TCs were presented in blue markers. The RI processes tend to occur to the northwest of their genesis, they shifted farther to the westward direction in May, November and December. Large scale environmental flow (e.g., vertical shear) and sea

surface temperature were believed to be possible inherent mechanisms (Holliday and Thompson, 1979; Kaplan and DeMaria, 2003; Wang and Zhou, 2008), which have significant impacts on the development of a TC and modulate its intensity since its genesis. Moreover, the occurrence latitude is another important climatological characteristic showing the seasonal variability of TCs. Kossin et al. (2016) investigated the TC activity using the metric, $\phi_{LMI}$ (the mean latitude where TCs reach their LMI), and reported a fairly uniform poleward migration of $\phi_{LMI}$ in the western North Pacific in past decades

since 1980. The seasonal discrepancy in latitude directly leads to differences in affected coasts each season. The universally observed westward locations (both TC genesis and RI on-set locations) of the landfalling RI TCs indicate that the RI TCs generated to the west of mean locations have higher probability to make landfalls afterwards and should be paid more attention.

### 4.2 Seasonal Variability of Landfalling *MACE* along the Coast

The TC numbers and associated *MACE* each month are plotted for the most hit coasts in the western North Pacific (Fig. 8) as in three representative regions (see definition in Fig. 2b). From north to south, both region 1 and region 2 were mainly affected by RI TCs from July to September, while region 3 were hit by active TCs mainly in October and November. Comparatively, non-RI TCs affected the southern Chinese coast in region 2 the most with absolute amounts of both TC numbers and energy, following by the Japanese coast in the second place. From Fig. 8c and Fig. 8f, the TCs carried greater

power and made landfalls in the Philippine coast were the ones have undergone RI processes. We further show the detailed distribution of *MACE* along the coast in most active TC months in Fig. 9. The Philippine coast, Taiwan coast and the





southern Japanese coast suffered from a higher level of TC attacks from the annual averaged statistics. In the terms of TC active season duration, it increases for coasts from north to south, especially for non-RI TCs. Moreover, it should note that compared with the explosive energy in October and November in the Philippine coast, the Vietnam & the southern Chinese

coast have suffered the longest (about 5 months) threats from considerable amounts of both TCs and associated energy.

## 5 Decadal Trends of RI

### 5.1 Variation in Occurrence Frequency of RI TCs

In Fig. 10a, the annual number of RI TCs does not show significant variation during the period 1986–2017. Owing to the minor trend of decrease in total TC numbers, the proportion of RI TCs among them demonstrates an increasing trend in the

past decades (Fig. 10b). Among the RI TCs, the landfalling ones show significant upward trend in both absolute number of occurred TCs and their percentage levels. The results indicate that the coast in the western North Pacific is exposed to an increasing probability of risk, being potentially attacked by more RI TCs in the future, which challenges an urgent requirement of enhancing the skills of TC intensity forecast, especially for the ones undergo RI just before landfall.

### 5.2 Variation in LMI of RI TCs

We plotted the LMI values of individual RI TCs in Fig. 11a and their subset - landfalling ones in Fig. 11b. From the calculated annual mean value of existing records, they both strengthen minorly but show insignificant trend. At the same time, the landfalling non-RI TCs imply to have a significant downward trend in LMI (Fig. 11c). Note that the RI TCs and non-RI TCs comprise the majorities of high intensity and low intensity TCs, respectively, thus their intensities upon landfalls tend to get stronger and weaker from our results. We further explore the RI TCs by three groups with different $\Delta V_{24}$ in Fig.

11d–f. A higher intensification rate is generally related to higher LMI, but meanwhile the LMI is also influenced by other factors, for instance, the intensification duration, the intensity when rapid intensification trigged, etc. Even that no significant trends are observed for RI TCs with different intensification rates, we notice increased occurrence of $\Delta V_{24} \geq 70 \ kt$ in recent ten years, which indicates potential occurrence of high intensity TCs owing to high intensification rate in the near future.

Note that it is not the only case in the western North Pacific that stronger TCs have strengthened and vice versa, previous

studies also yield similar trends in both the Northern and the Southern Hemisphere (Song et al., 2018). LMI variation of RI TCs and non-RI TCs hence definitely would vary the LMI distribution of TCs, which is shown in Fig. 12a. Compared with period 2001–1986, the LMI of TCs in period 2002–2017 migrates to higher intensities in both ends of LMI distribution, while to lower intensities for TCs with intensity range from 45 kt to 115 kt. The changes in LMI distribution of RI TCs and non-RI TCs (Fig. 12b-c) well explained the intensity shift in LMI distribution: a positive intensity change of RI TCs around

125 kt is mainly responsible for the strengthened strong TCs in the past decades; larger negative changes at 45 kt, 75 kt and 95 kt demonstrate the trend of weakened moderate TCs.



### 5.3 Variation in Cyclone Energy of Landfalling TCs

Apart from the LMI which describes the maximum intensity during a TC's lifespan, the metric *MACE* quantifying the local influence on coast has been utilized to obtain its annual mean value (period 1986–2017) of landfalling TCs in Section 3.3,

here we further calculate the anomalies by comparing the average cyclone energy during period 1986–2001 and period 2002–2017 with it (Fig.12). The discrepancies in trend of regional patterns could be observed from the decadal variation in spatial distribution of the landfalling TCs. Compared with standard distribution during period 1986–2017 (Fig. 5d for the RI TCs), the southern Chinese coast and the coast along the Korean Peninsula have gained increased landfalling cyclone energy in recent decades affected by RI TCs, while other coasts mainly showing spatial heterogeneity where the northern Philippine

coast underwent larger variation at level of tropical storm (Fig. 13b). For the non-RI TCs, the decline of landfalling cyclone energy is easily observed in most coasts, including the Philippine coast, the southern Japanese coast, and the Vietnam coast. It implies a generally universal decrease pattern of non-RI TCs during period 2002–2017 (Fig. 13d).

Considering the regional differences of landfalling cyclone energy, we plot summation of *MACE* for separate 3 regions defined in Fig. 2b. The varying amplitude of the summated cyclone energy of the landfalling RI TCs in Fig. 14a–c illustrates

that the Philippine coast in region 3 has been battered with the most power in past decades, while the annually averaged values in region 1 and 2 are about 1/2 and minorly smaller than that of the former. Statistically small numbers (2–3) of RI TCs have made landfalls in these 3 regions, which made the cyclone energy being correlated quite tightly with TC numbers to a great degree. This correlation becomes much weaker in the case of the landfalling non-RI TCs (Fig. 14e–g). In terms of the cyclone energy magnitude level, the Vietnam & the southern Chinese coast in region 2 has been exposed to comparable

power induced by RI TCs and non-RI TCs, while the landfalling cyclone energy brought to other coasts by non-RI TCs was much smaller than that of RI TCs, and tend to become weaker from the significant downward trend in Fig. 14e and Fig. 14g. As a whole, the landfalling non-RI TCs show significant decrease trend of cyclone energy in the western North Pacific; meanwhile, the power of RI TCs, which is almost two times of the former, tend to increase in the near future. This upward trend of RI TC energy remind us that TCs with extremely high intensity should be paid more attention along the Philippine

coast, at the same time the southern Chinese coast should prepare positively for both frequent occurrence and potential high intensity TCs.

### 6 Conclusions

RI is an essential phenomenon that accompanies with TCs' development and occurred in the majority of the major storms. It was highly correlated with LMI of TCs and affected associated landfalling cyclone energy, which could cause potential

devastating damage to coastal communities. In the present study, the distribution of historical landfalls in the western North Pacific during 1986–2017 was explored and the seasonal and decadal trend of related characteristics (frequency, intensity, and landfalling energy, etc.) were investigated for RI TCs and non-RI TCs, respectively, which are two representative types





of TCs characterized with different intensification rates. A metric, $MACE$, was proposed to depict spatial distribution of cyclone energy prior to landfalls.

In the western North Pacific basin, 31.2 % TCs have underwent at least one RI processes and 341 of them made landfalls after RI from 946 historical TCs during 1986–2017, utilizing the definition of 95[th] percentile from the accumulative probability distribution of over-water 24-h TC intensity change, $\Delta V_{24}$. On average, annually 9 TCs underwent RI, and RI occurrence is concentrated in sea areas to the east of Philippines, namely the region 10˚ N–20˚ N, 124˚ E–137˚ E. The mean genesis and on-set locations of landfalling RI TCs are found westward compared with the non-RI TCs. The Philippine coast,

the southern Chinese coast and the coast along the southern Japan are the three main areas affected by RI TCs. The southern Chinese and Japanese coast have roundly 2–3 landfalls by RI TCs each year during the major active typhoon season (July–September), while the Philippine coast was affected mainly in the late autumn with almost 2 times of cyclone energy when TCs made their landfalls. Although the RI TC numbers are averagely half of that of the non-RI TCs, their landfalling energy is around 2 times of the latter. Compared with the declined trend of annual TCs numbers, the absolute number and the ratio

of RI TCs that made landfalls have a significant upward trend during this period, with insignificant variation in the LMI. The spatial distribution of cyclone energy anomalies illustrate that the power from the landfalling RI TCs increased especially along the coast along the southern China and the Korean Peninsula. We attribute this regional variation to the poleward migration of $\phi_{LMI}$ in the western North Pacific, which also influenced the distribution of landfalling intensity. Meanwhile, from both the overall and regional patterns, the LMI and landfalling cyclone energy by non-RI TCs in the western North

Pacific indicate downward trends in the past decades. The above trend of RI TCs and non-RI TCs in the past decades led to changes of the LMI distribution in the western North Pacific, indicating strong TCs strengthened mainly due to increased RI and weak TCs have gotten weaker.

The trend of landfalling cyclone energy pose a direct potential risk to coastal communities, especially form the RI TCs. We believe the knowledge of landfalling patterns by those two types of TCs and their trends could assist as data basis to

establish relationship between landfalling TC intensities and local damages, which are also correlated with societal factors, and help to better plan for the regional counter measures in the western North Pacific basin.

**Appendix A Conversion of RI Definition from Best Track Data of JTWC, CMA and JMA**

The averaging periods for MSW differ among different agencies, namely the JTWC, CMA (China Meteorological Administration) and JMA (Japan Meteorological Agency) use 1-min, 2-min and 10-min MSW, respectively. Note that RI

definition uses the 1-min MSW, which implies the best track data provided in other averaging periods should be converted before usage. Even that comparison of results from different best track data sets is beyond the scope of the present paper, this section intends to provide an option of RI definition when using best track data with different averaging periods of MSW. Harper et al. (2010) suggested guideline of converting wind speed of TC over land and over water, which was adopted in many previous studies, but the reasonability of using a uniform conversion coefficient for all wind speeds may still need



more study. Kang and Elsner (2019) suggested a quantile approach which matches two data sources at the same probability level from the accumulative probability distribution of wind speed. In practice, the high tail in the accumulative probability distribution gives much more uncertainty to assess the conversion coefficient between the 1-min and 2-min/10-min MSW. Therefore, we utilize the original raw 2-min/10-min MSW instead, but matching intensity change directly at the same percentiles from the accumulative probability distribution of $\Delta V_{24}$ from 1-min MSW. For the case of the western North

Pacific, we use the 95[th] percentile for uniform definition, and have the estimated results listed in Table A1.

**Author Contribution**

JY designed the study and performed the analysis from IBTrACS data set. JY and MC generated the plots. JY prepared the manuscript with contributions from all other co-authors. All authors contributed to the discussion and interpretation of the results.

**Acknowledgements**

This research is supported by the National Key Research and Development Program of China (Grant No. 2018YFC0407506). This work is also partially supported by the Key Laboratory of Coastal Disaster and Defence of Ministry of Education, Hohai University.

**Data Sets**

The version v04r00 of IBTrACS data set (Knapp et al., 2010) was retrieved via https://www.ncdc.noaa.gov/ibtracs/index.php?name=ib-v4-access in March 2019 and still available from this link.

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





**Table 1.** Definitions of TC groups with different intensification rate in 1-min MSW during 1986–2017 in the western North Pacific. $\Delta V_{24}^{max}$ represents lifetime maximum of 24-h intensity change $\Delta V_{24}$ for a TC, and always has a positive value.

| TC intensification group | Criterion |
|---|---|
| Rapid intensification (RI) | $\Delta V_{24}^{max} \geq 35\ kt$ |
| Slow intensification (SI) | $10\ kt \leq \Delta V_{24}^{max} < 35\ kt$ |
| Neutral (N) | $\Delta V_{24}^{max} < 10\ kt$ |

**Table 2.** Illustration of cyclone energy counts in four basic grades. Basic $CE$ is determined utilizing the lower limit of each intensity grade.

| Grade | Illustration | $MACE$ Unit |
|---|---|---|
| I | Equivalent to tropical depression | $CE_1 < 34^2$ |
| II | Equivalent to tropical storm | $CE_2 = 34^2$ |
| III | Equivalent to Category 1 storm and above (moderate intensity storm) | $CE_3 = 64^2$ |
| IV | Equivalent to Category 4 storm and above (high intensity storm) | $CE_4 = 113^2$ |

**Table A1.** Criteria of 24-h intensity change in MSW with different averaging periods by matching at the same level of accumulative density of 24-h intensity change 1-min MSW ($\Delta V_{24}$ in this text). The criteria in table are obtained for the western North Pacific basin from same TC samples of the IBTrACS best track data during 1986–2017.

| Percentile | Criterion | | |
|---|---|---|---|
| | JTWC (1-min) | CMA (2-min) | JMA (10-min) |
| 58.2 % | $10\ kt$ | $9\ kt$ | $5.5\ kt$ |
| 94.7 % | $35\ kt$ | $29\ kt$ | $25.5\ kt$ |
| 98.5% | $50\ kt$ | $39\ kt$ | $35\ kt$ |
| 99.8 % | $70\ kt$ | $50\ kt$ | $45\ kt$ |




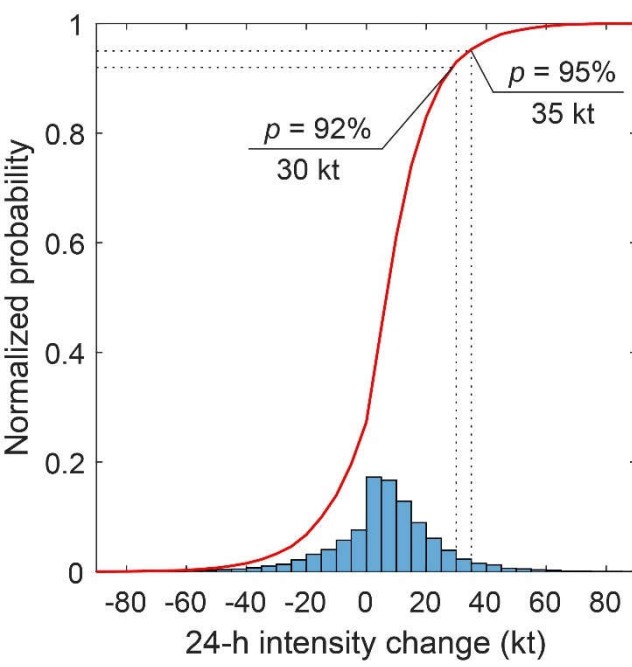

**Figure 1.** Normalized probability density distribution and cumulative distribution of 24-h intensity change $\Delta V_{24}$ in the western North Pacific using IBTrACS-JTWC data during 1986–2017. The probability levels associated with two most popular criteria of intensification rates are marked on the distribution curve.

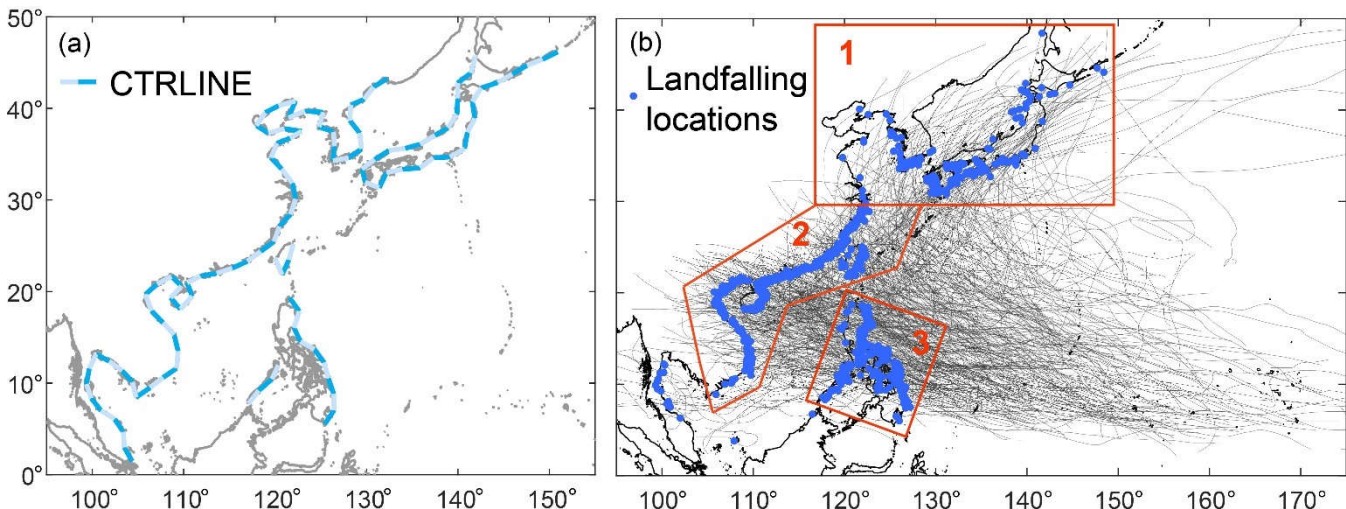

**Figure 2.** (a) Distribution of control line segments CTRLINE (marked by alternating blue lines and light blue lines) and (b) Landfalling TC tracks and associated landfall locations (blue dots) in the western North Pacific during 1986–2017. Three subregions are illustrated with red polygons in (b).




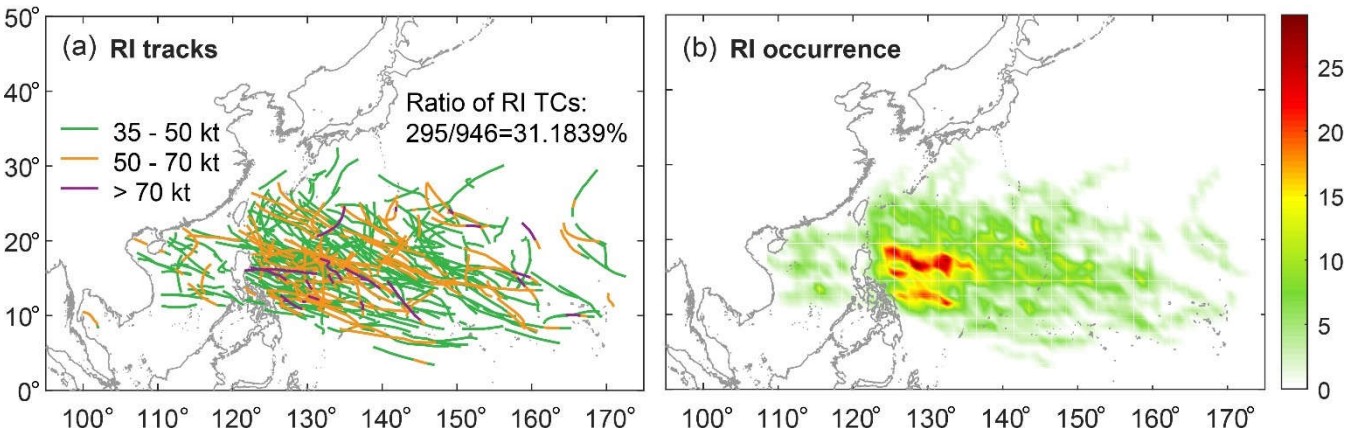

**Figure 3.** (a) TC track segments underwent RI and (b) corresponding RI occurrence density calculated at 1°×1° grid in the western North Pacific from the IBTrACS-JTWC data during period 1986–2017. Three grades of intensity change $\Delta V_{24}$ are distinguished using different colors in (a).

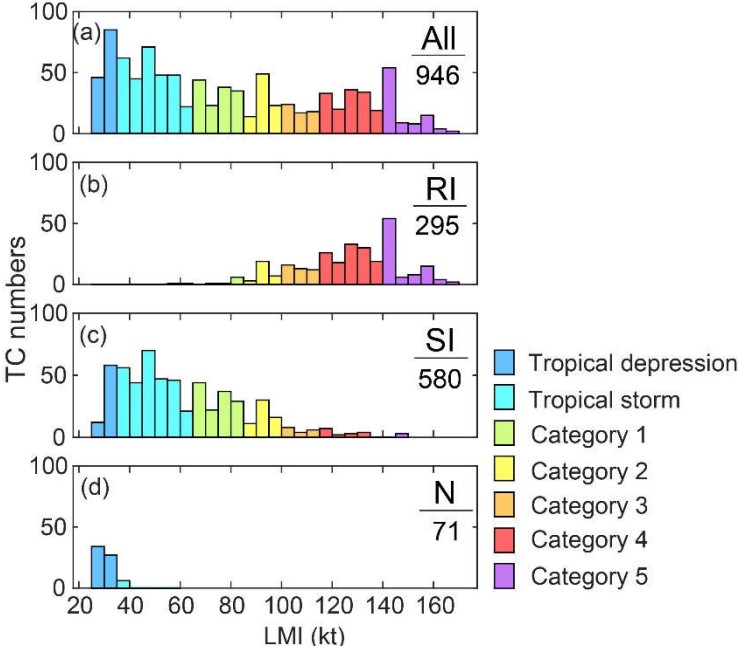

**Figure 4.** (a) LMI distribution of TCs in the western North Pacific during period 1986–2017. (b–d) LMI distribution in three intensification groups. Total TC numbers of each group are marked in corresponding subfigures. TC LMI is determined using the 1-min MSW from IBTrACS-JTWC data. The plots are generated in 5 kt-binned bars and different face colors represent TC intensity categories on the Saffir–Simpson wind scale.



**Figure 5.** Landfall locations and landfalling intensities of RI TCs (a) and non-RI TCs (b) during 1986–2017 in the western North Pacific (data source: IBTrACS-JTWC). (c) The distribution of annually averaged landfalling cyclone energy along the CTRLINE from all landfalling TCs and that of RI TCs (d) and non-RI TCs (e), respectively, associated with (a) and (b) utilizing the same period of data. The cyclone energy is represented in 1–n counts for 4 grades of intensity (see Table 2).




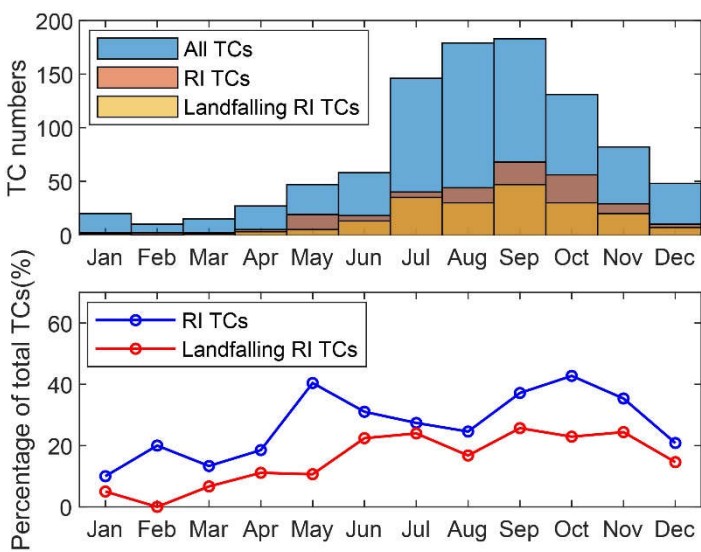

**Figure 6.** (a) Seasonal statistics of TC numbers that underwent RI and made landfalls after RI processes and (b) their corresponding percentages, from TC records in the western North Pacific during 1986–2017 (Data source: IBTrACS-JTWC).

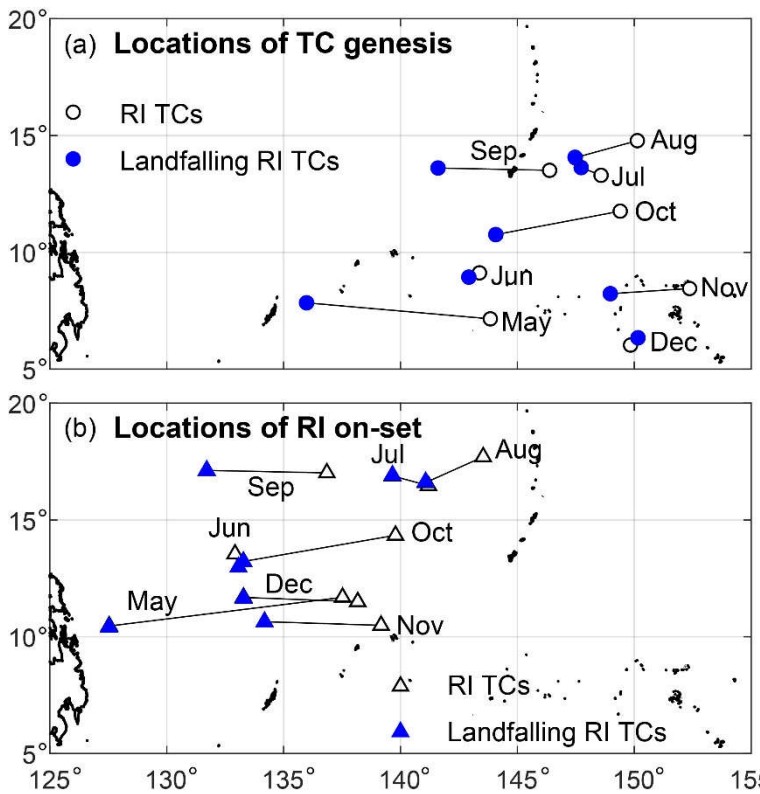

400

**Figure 7.** (a) Mean locations of TC genesis that underwent RI (white dots) and made landfalls after at least one RI (blue dots), a black line connects each pair of locations occur at the same month. (b) Similar as (a), but the mean locations of RI on-set for TCs that underwent RI (white triangles) and made landfalls after at least one RI (blue triangles). If a TC underwent more than one RI during its lifespan, only the first RI on-set location is considered in the calculation.

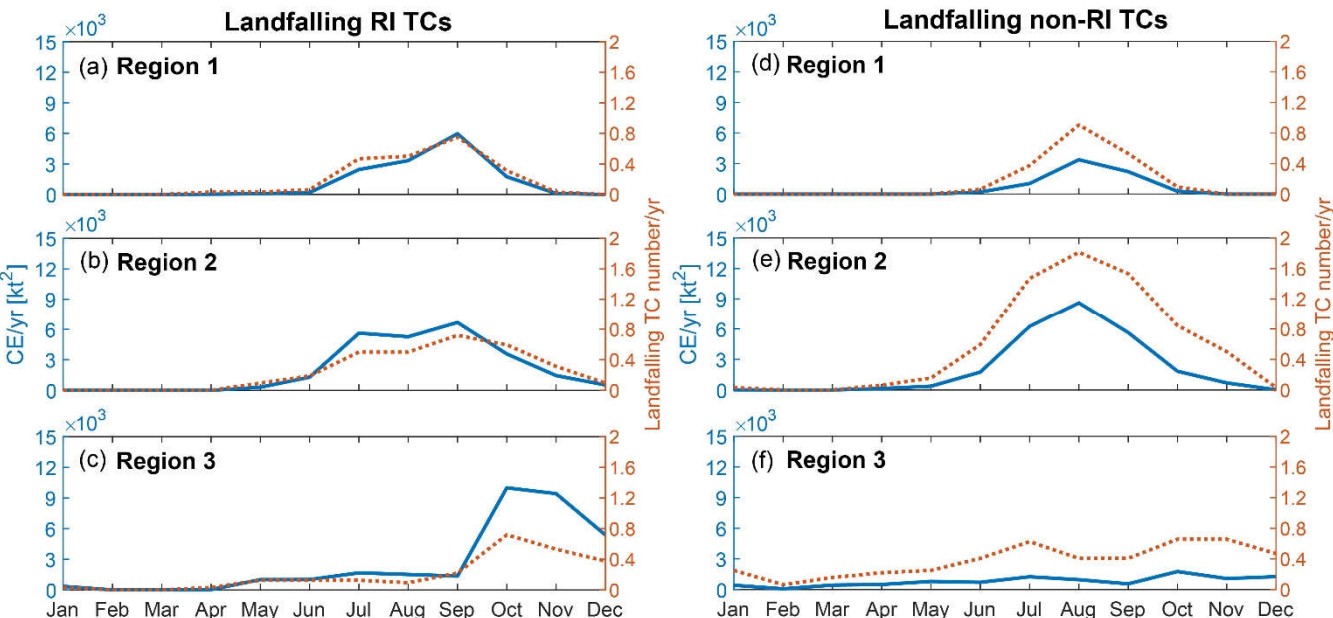

**Figure 8.** Seasonal variation of annually averaged *MACE* and associated TC numbers in representative regions 1-3 for the landfalling (a–c) RI TCs and (d–f) Non-RI TCs during the period 1986–2017.

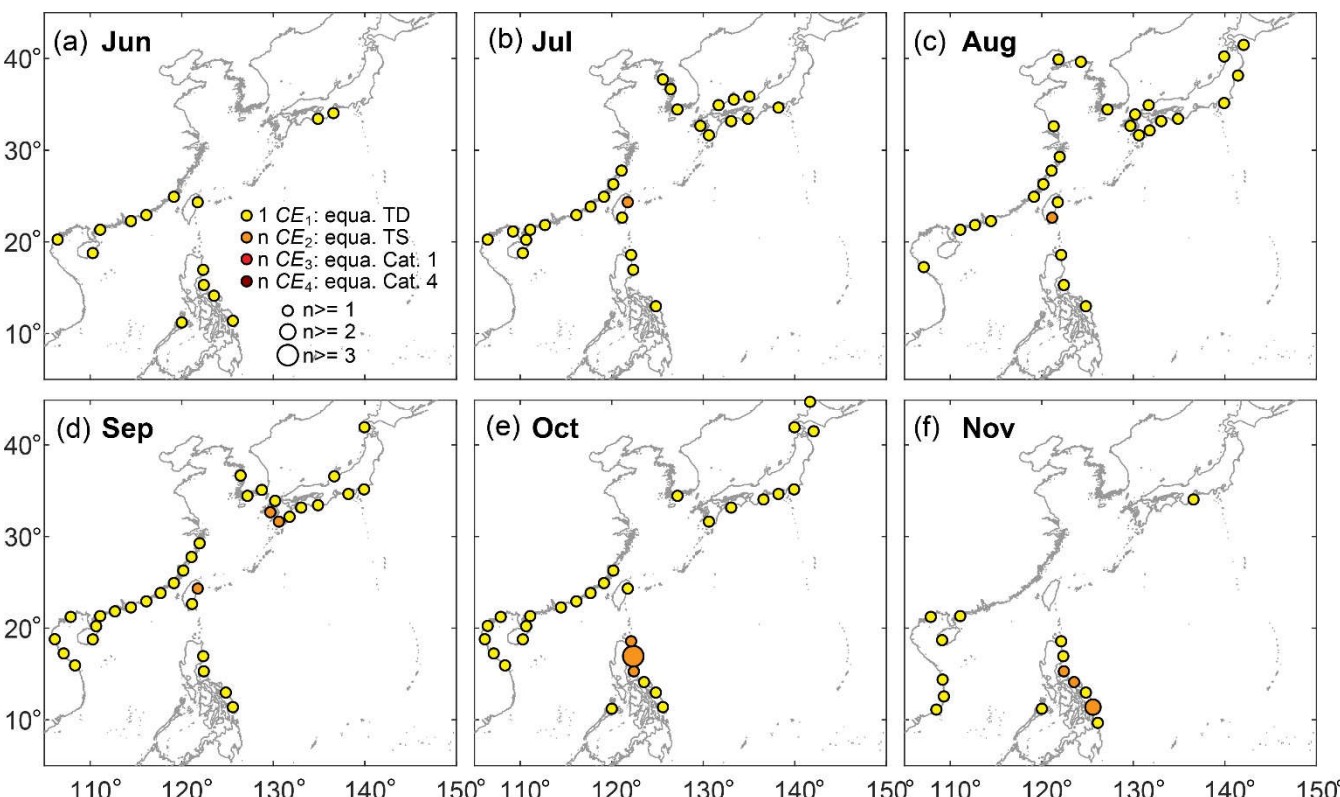




**Figure 9.** Distribution of annual averaged landfalling cyclone energy (1986–2017) along the coast of the western North Pacific are plotted for selected relative TC active months.

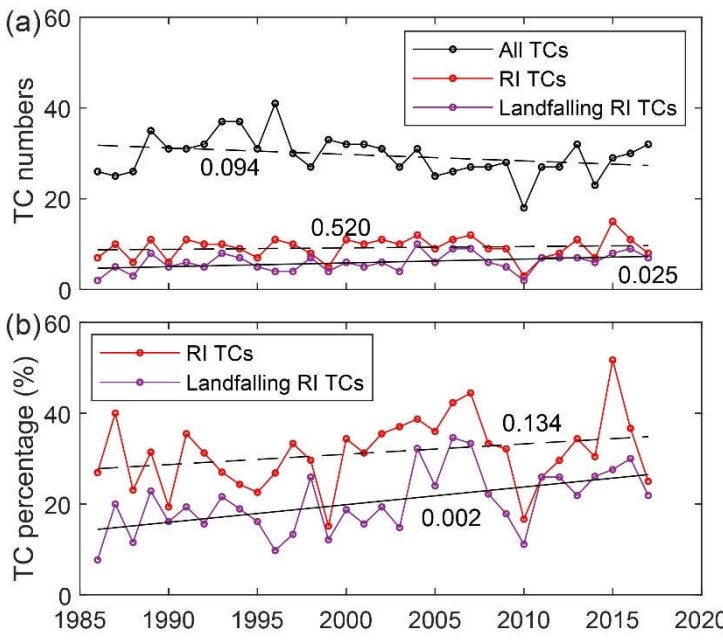

**Figure 10.** Time series of TC numbers that underwent RI and the landfalling RI TCs (above panel) and their corresponding percentages (bottom panel), from total TC records in the western North Pacific during 1986–2017 (Data source: IBTrACS-JTWC). Straight lines are linear regressions of time series and are solid if statistically significant (<0.05), associated values of significance are marked near the linear regression trend.

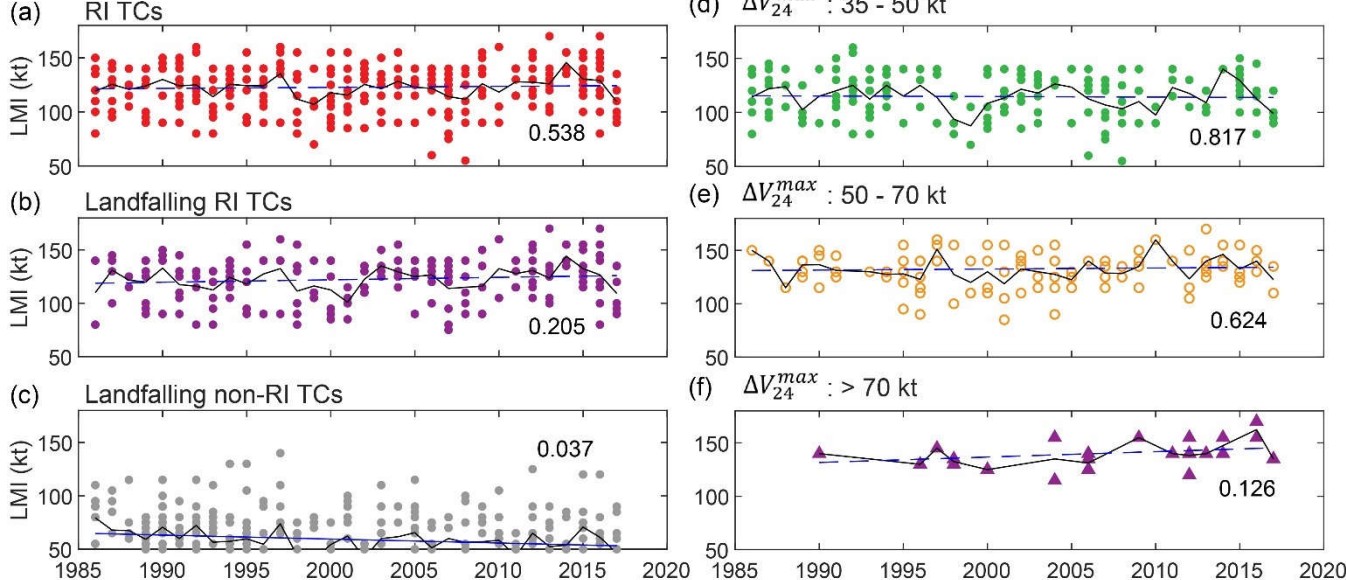

**Figure 11.** Variation of LMI by presenting within categories of (a) RI TCs and (b–c) Landfalling TCs during 1986–2017 in the western North Pacific; the former with different RI rates are separately plotted in subfigure (d–f). The blue straight lines are linear regressions of

420 annual LMI mean value (black curves) and are solid if statistically significant (<0.05), associated values of significance are marked near the linear regression trend.

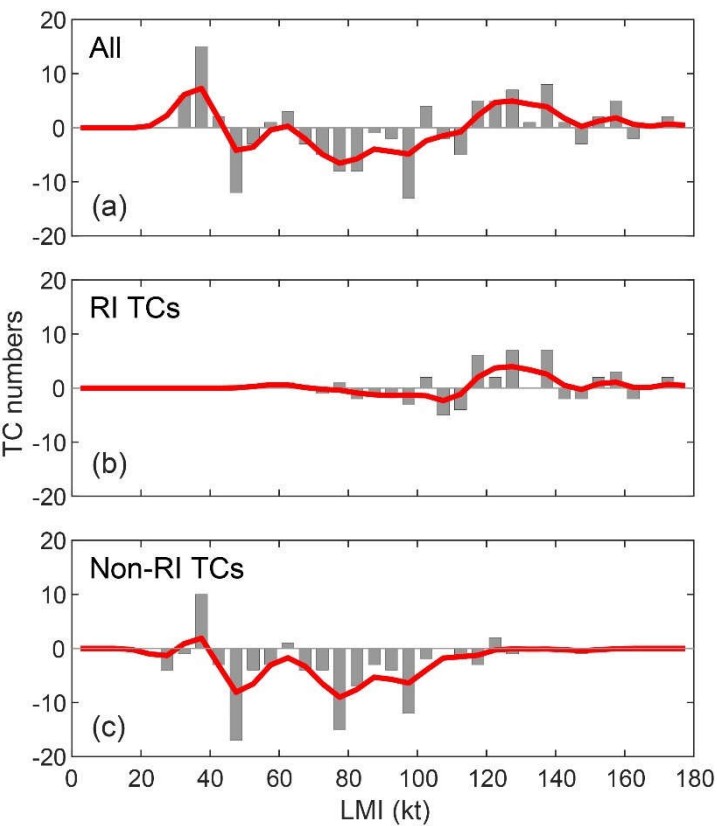

**Figure 12.** The variation of LMI distribution (period 2002–2017 minus period 2001–1986) for (a) All TCs; (b) RI TCs and (c) Non-RI TCs in the western North Pacific. The plots are generated in 5 kt-binned grey bars and the red curves are smoothed by a 5-point low-pass Gaussian filter.
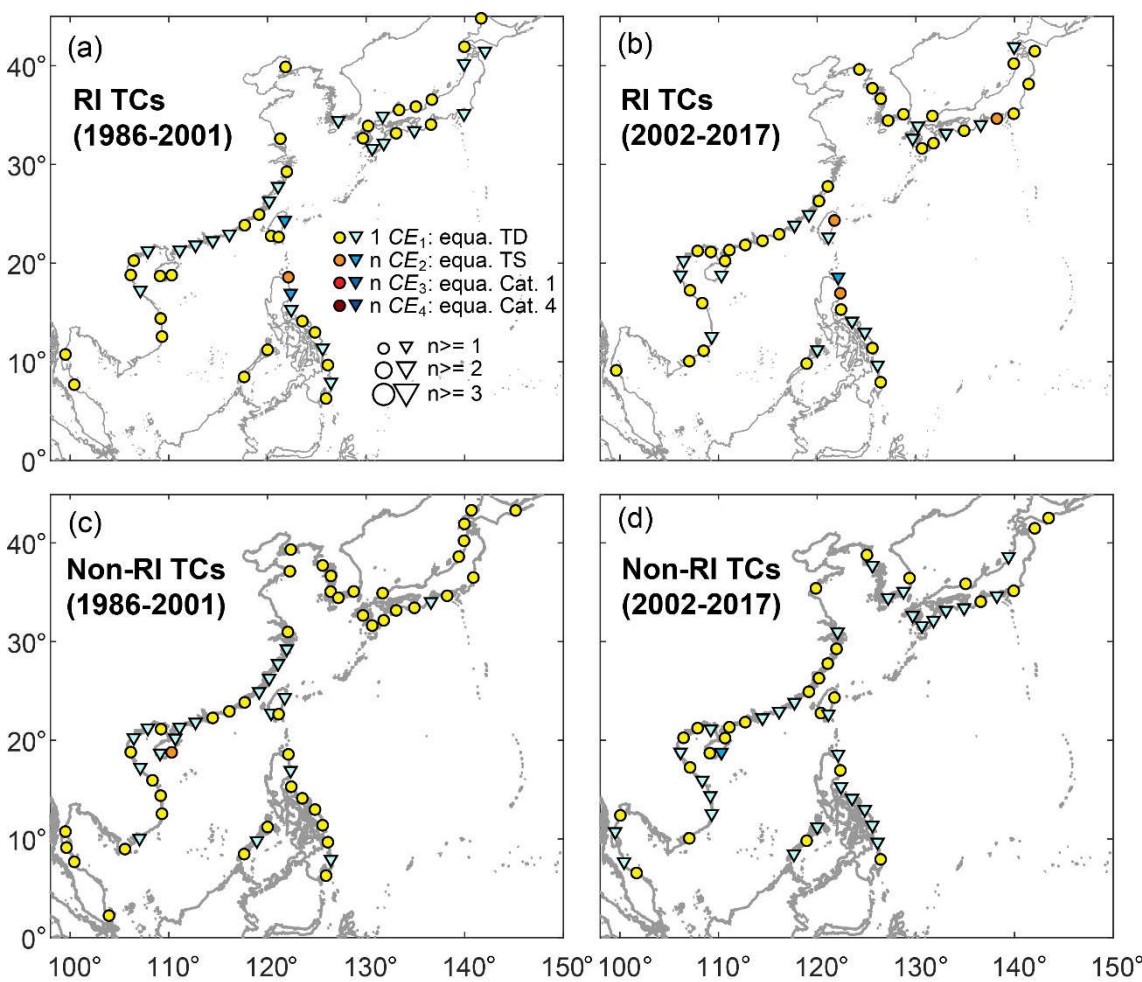

425

**Figure 13.** The anomalies of annually averaged landfalling cyclone energy along the CTRLINE from (a–b) RI TCs and (c–d) non-RI TCs are calculated during period 1986–2001 (left panel) and period 2002–2017 (right panel), respectively. Circles and triangles denote positive and negative variation of cyclone energy, which are relative to mean values during period 1986–2017 in Figure 5d-e.




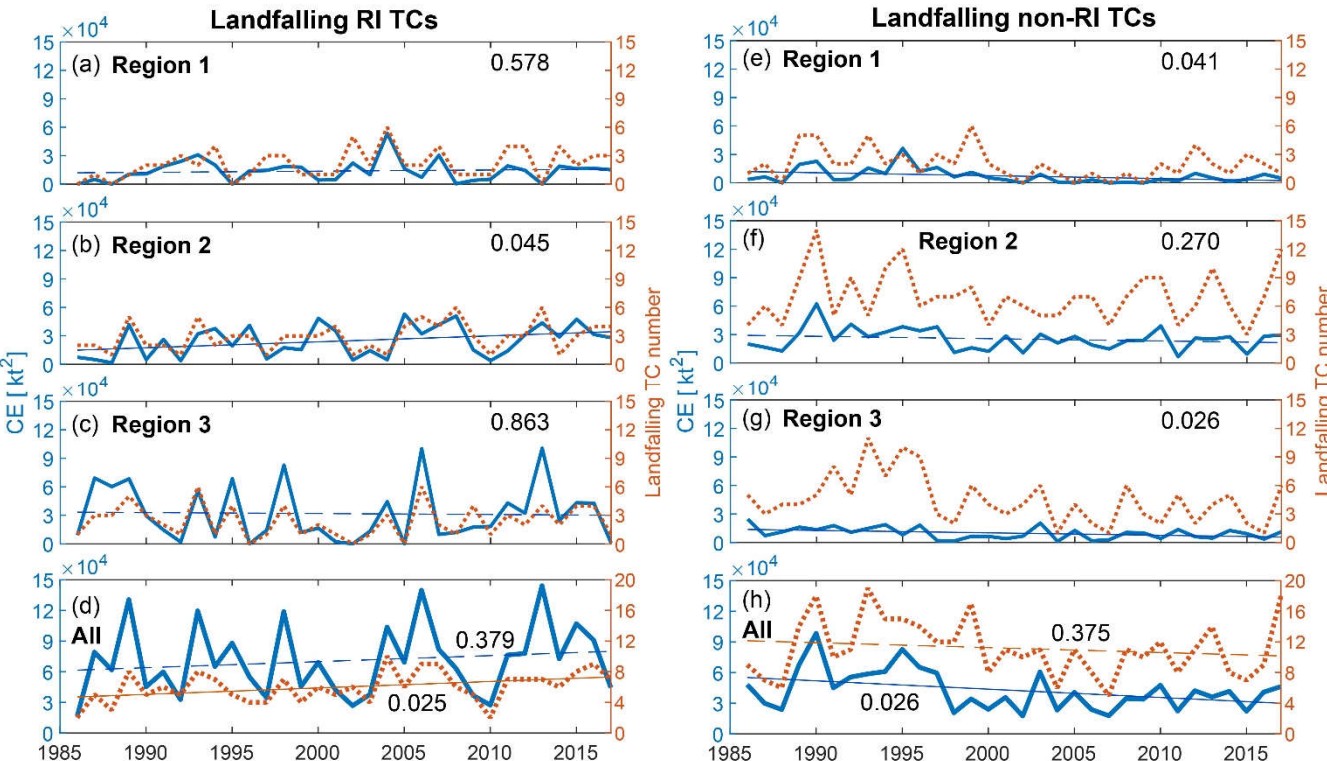

**Figure 14.** Decadal variation of summated *MACE* associated with the landfalling (a–d) RI TCs and (e–h) Non-RI TCs are presented for representative regions 1-3 and along all affected coast in the western North Pacific. The total energy and total TC numbers are distinguished by blue and orange colors. The straight lines are linear regressions using corresponding colors, they are solid if statistically significant (<0.05), associated values of significance are marked near the linear regression trend.