# Peer review of "Landfalls of Tropical Cyclones with Rapid Intensification in the Western North Pacific"

_Natural Hazards and Earth System Sciences, 2019_

## Referee Comment (RC1) · Tso-Ren Wu (Referee) · 17 Nov 2019

This paper investigates the phenomena of rapid intensification (RI) of landfalls of tropical cyclones (TC). The topic is interesting, and the results are convincing to me. The statistical results show that about 1/3 of the TCs underwent RI processes, and about 1/3 of historical TCs made landfalls after RI. This paper also found that the regions of the southern Chinese coast and southern Japan have increased the trend of cyclone energy since 1986. I am pleased to read this paper. Some of the questions and comments are listed below: 1. It is really difficult to understand the sentences in the abstract, "The frequent-occurrence region of RI is found in sea areas to the east of Philippines, and the mean genesis and on-set locations of landfalling TCs underwent RI had westward components compared with the ones did not make landfalls." A revise

is suggested. 2. In the abstract, "The coasts in the latter two regions have increased trend of cyclone energy since 1986, which possibly correlates with the poleward migration of the mean latitude where TCs reach their lifetime maximum intensities (LMI)." Because no correlation analysis in the context, it is not proper to put this "possible" guessing in the abstract. However, detailed correlation analysis is highly expected to be done in this paper or in the future works. 3. Figure 11 shows the trend of RI TCs, Landfalling RI TCs, and Landfalling non-RI RCs. However, adding the trend of All TCs is required to understand the correlation between RI TCs and All TCs. This can be used to explain if the increasing trends of RI TCs and Landfalling RI TCs are directly related to the All TCs or not.

---

## Author Comment (AC1) · 28 Nov 2019

We thank reviewer #1 for the valuable suggestions. In the revised manuscript, we have made changes according to the suggestions and comments and highlighted where those changes are made. The point-by-point replies to the comments are presented below. General comments: This paper investigates the phenomena of rapid intensification (RI) of landfalls of tropical cyclones (TC). The topic is interesting, and the results are convincing to me. The statistical results show that about 1/3 of the TCs underwent RI processes, and about 1/3 of historical TCs made landfalls after RI. This paper also found that the regions of the southern Chinese coast and southern Japan have increased the trend of cyclone energy since 1986. I am pleased to read this paper. Some of the questions and comments are listed below. Author's response: Thank you

for your interest in this topic and your encouraging comments. We were also impressed that RI plays an important role influencing cyclone intensity and energy. We executed extra analysis according to your suggestions and the corresponding changes in the revised manuscript will be shown in following detailed responses.

Comment #1. It is really difficult to understand the sentences in the abstract, "The frequent-occurrence region of RI is found in sea areas to the east of Philippines, and the mean genesis and on-set locations of landfalling TCs underwent RI had westward components compared with the ones did not make landfalls." A revise is suggested. Author's response: Thanks for the comment. To avoid confusion, we changed the sentence. Revisions in manuscript: The original sentence has been replaced with "The frequent-occurrence region of RI is found in sea areas to the east of Philippines, and the mean genesis and on-set locations of landfalling TCs underwent RI shifted to more western regions compared with the ones did not make landfalls.".

Comment #2. In the abstract, "The coasts in the latter two regions have increased trend of cyclone energy since 1986, which possibly correlates with the poleward migration of the mean latitude where TCs reach their lifetime maximum intensities (LMI)." Because no correlation analysis in the context, it is not proper to put this "possible" guessing in the abstract. However, detailed correlation analysis is highly expected to be done in this paper or in the future works. Author's response: Thanks a lot for the suggestions. We came into this conclusion in view of the finding by Kossin et al. (2016), by whom an uniform poleward migration of the mean latitude where TCs reach their LMI in the western North Pacific in past decades since 1980 was reported. To confirm the correlation between the increased cyclone energy in the coasts of southern China and southern Japan and the poleward migration of the mean LMI latitude, we analyse the trend of the mean LMI and the mean latitude where TCs reach their LMI each year in the western North Pacific during 1986–2017, for all TCs and RI TCs, respectively. From Fig. 1b, we obtain similar conclusion as Kossin et al. (2016) that mean latitude where TCs reach their LMI has migrated poleward, the magnitude was about $1°$ in

the period 1986–2017; more than that, the migration rate of RI TCs show significant trend and was up to 2.5°. The migration of mean latitude where TCs reach their LMI in the western North Pacific towards North Pole's direction could explain the increased cyclone energy in both regions of southern Chinese coasts and southern Japanese coasts. Revisions in manuscript: In the abstract, the above-mentioned sentence has been replaced with "The coasts in the latter two regions have increased trend of cyclone energy since 1986, which correlates with the poleward migration of the mean latitude where TCs reach their lifetime maximum intensities (LMI)." We added Fig. 1 as Figure 11 in the revised manuscript. In Line 192, the title of section 5.1 "Variation in Occurrence Frequency of RI TCs" has been replaced with "Variation in Occurrence Frequency of RI TCs and their LMI. From Line 199, we added sentences "In Fig. 11, we further show the trend of the mean LMI and the mean latitude where TCs reach their LMI (ïĄę_LMI) each year in the western North Pacific during 1986–2017, for all TCs and RI TCs, respectively. The upward trend of mean LMI of RI TCs in Fig. 11a is insignificant and yields to an opposite trend for all TCs, which illustrates that the decreased intensities of non-RI TCs in the past decades played a more important role. Using the metric ïĄę_LMI proposed by Kossin et al. (2016), we found ïĄę_LMI of all TCs in the western North Pacific basin has migrated poleward about 1° in the period 1986–2017 (Fig. 11b), moreover, the migration rate of ïĄę_LMI owing to RI TCs show a significant trend and was up to 2.5° in magnitude in the past decades. The migration trend of ïĄę_LMI reveals that the cyclone activities have shifted to northern regions gradually and tend to increase associate risks there. " as a new paragraph.

Comment #3. Figure 11 shows the trend of RI TCs, Landfalling RI TCs, and Landfalling non-RI RCs. However, adding the trend of All TCs is required to understand the correlation between RI TCs and All TCs. This can be used to explain if the increasing trends of RI TCs and Landfalling RI TCs are directly related to the All TCs or not. Author's response: Thanks for the suggestion. We have added the trend of all TCs in Fig. 1a, to better illustrate the trend comparison of different categories of TCs, the trend of RI TCs has been included too. The opposite trend of all TCs and RI TCs illustrates that the

downward trend of all TCs was mainly influenced by non-RI TCs as their intensities decreased in the past decades. Revisions in manuscript: See our response to Comment #2.

References Kossin, J. P., Emanuel, K. A., & Camargo, S. J. (2016). Past and Projected Changes in Western North Pacific Tropical Cyclone Exposure. Journal of Climate, 29(16), 5725–5739. https://doi.org/10.1175/JCLI-D-16-0076.1

[Figure]

**Figure 11**. Time series of (a) the mean LMI and (b) $\phi_{LMI}$, for all TCs (blue) and RI TCs (red) in the western North Pacific during 1986–2017. (Data source: IBTrACS-JTWC). Straight lines are linear regressions of time series and are solid if statistically significant (<0.05), associated values of significance are marked at the end of linear regression trend with same colors. Shadings show 95% confidence bounds, associated migration rates are marked as well as their 95% confidence intervals.

**Fig. 1.** Time series of (a) the mean LMI and (b) the mean latitude where TCs reach their LMI, for all TCs (blue) and RI TCs (red) in the western North Pacific during 1986–2017.

[Figure]

---

## Referee Comment (RC2) · Anonymous Referee #2 · 6 Feb 2020

The manuscript in its current form is too poorly written to be properly evaluated. I find myself spending 90% of my time simply trying to interpret what the authors actually mean from a language standpoint rather than interpreting the science.

I suggest having a native English speaker work with the authors (and maybe a translator) to substantially re-write the manuscript.

I think there could be some worthwhile scientific information in the article, but it's too poorly written and organized to make that determination currently.

---

## Author Comment (AC2) · 28 Feb 2020

We thank reviewer #2 for the constructive suggestions.

General comments: The manuscript in its current form is too poorly written to be properly evaluated. I find myself spending 90% of my time simply trying to interpret what the authors actually mean from a language standpoint rather than interpreting the science. I suggest having a native English speaker work with the authors (and maybe a translator) to substantially re-write the manuscript. I think there could be some worthwhile scientific information in the article, but it's too poorly written and organized to make that determination currently.

Author's response: We have made great effort to improve the language issue raised

by Reviewer #2. This is evidenced by the heavy edits in the revised version. Both the logical flow and language are significantly improved. After several rounds of edits, we also consulted with professional language services for further improvement. We believe the scientific information is much clearer in this version and look forward to your comments.

Author's change to manuscript: We listed highlighted changes below. Please kindly refer to the revised manuscript later. It is not easy to list all the changes here since the edits are numerous. After editor's decision, we would upload both the correction-tracking and clean version of the revised manuscript following corresponding instructions. Highlighted changes in the revised manuscript: 1) We had adjusted the structure to have a better logical flow; 2) The abstract had been shortened and reorganized to fulfil the requirement of character limitation; 3) We improved the language for the whole manuscript; 4) We corrected the figures by adding the notations of north arrow and scale in the plots.